

# Phosphodiesterase 4 inhibition as a novel treatment for stroke

Jiahong Zhong[1], Xihui Yu[2] and Zhuomiao Lin[1]

[1] Department of Clinical Pharmacy, Meizhou People's Hospital, Meizhou, Guangdong, China
[2] Department of Pharmacy, The Second Affiliated Hospital of Shantou University Medical College, Shantou, Guangdong, China

## ABSTRACT

The incidence of stroke ranks third among the leading causes of mortality worldwide. It has the characteristics of high morbidity, high disability rate and high recurrence rate. The current risk associated with stroke surgery is exceedingly high. It may potentially outweigh the benefits and fail to ameliorate the cerebral tissue damage following ischemia. Therefore, pharmacological intervention assumes paramount importance. The use of thrombolytic drugs is most common in the treatment of stroke; however, its efficacy is limited due to its time-sensitive nature and propensity for increased bleeding. Over the past few years, the treatment of stroke has witnessed a surge in interest towards neuroprotective drugs that possess the potential to enhance neurological function. The PDE4D gene has been demonstrated to have a positive correlation with the risk of ischemic stroke. Additionally, the utilization of phosphodiesterase 4 inhibitors can enhance synaptic plasticity within the neural circuitry, regulate cellular metabolism, and prevent secondary brain injury caused by impaired blood flow. These mechanisms collectively facilitate the recovery of functional neurons, thereby serving as potential therapeutic interventions. Therefore, the comprehensive investigation of phosphodiesterase 4 as an innovative pharmacological target for stroke injury provides valuable insights into the development of therapeutic interventions in stroke treatment. This review is intended for, but not limited to, pharmacological researchers, drug target researchers, neurologists, neuromedical researchers, and behavioral scientists.

## INTRODUCTION

Cerebral stroke, also known as stroke, cerebral vascular accident (CVA), is an acute cerebrovascular disease (*Krishnamurthi et al., 2013*; *Feigin et al., 2014*; *Diener et al., 2022*). Cerebral stroke results from the interruption of blood supply, which is typically due to the rupture of blood vessels or the obstruction of blood vessels by blood clots. This leads to the cessation of oxygen and nutrient supply, subsequently causing irreversible damage to brain tissue in the form of hypoxia and ischemic necrosis (*Balami, Chen & Buchan, 2013*; *Dunbar & Kirton, 2018*; *Potter, Tannous & Vahidy, 2022*; *Qin et al., 2022*). Stroke is characterized by "three highs and one low", namely high incidence, high recurrence rate, high disability rate and low cure rate. It is the third leading cause of death in the world (*Chen et al., 2017b*; *Campbell et al., 2019*; *Kuriakose & Xiao, 2020*). The standard

Corresponding author
Zhuomiao Lin,
linzhuomiao@mzrmyy.com

prevalence of stroke in people over 65 years old is about 5% to 7%, and ischemic stroke accounts for 80% of all stroke patients (*Krishnamurthi et al., 2013*; *Hankey, 2017*; *Saini, Guada & Yavagal, 2021*). Over the past few decades, the incidence of stroke has exhibited an increasing tendency in developing countries, whereas a decreasing trend has been observed in developed countries. Moreover, the secondary complications ensuing from stroke pose a significant threat to the overall health and well-being of the population (*Reeves et al., 2008*; *Barthels & Das, 2020*). Currently, prevention strategies can effectively reduce the disease burden associated with stroke, with particular emphasis on the early and proactive primary prevention of stroke risk factors. Surgical treatment and drug treatment are the two main treatment strategies for stroke patients. However, surgical treatment requires patients with certain disease characteristics and surgical conditions, and the risks and benefits of surgery are difficult to predict (*Hankey, 2017*; *Li et al., 2017b*; *Prust, Forman & Ovbiagele, 2024*).

Alternatively, thrombolytic drugs are currently the most commonly used drugs for the treatment of stroke in the clinic and are the most effective treatment for the acute phase of ischemic stroke (*Merkler et al., 2017*; *Greco et al., 2023*). However, the time dependence of thrombolytic drugs and the characteristics of increased hemorrhagic transformation limit their use (*Phipps & Cronin, 2020*; *Greco et al., 2023*; *Tsivgoulis et al., 2023*). The ischemic penumbra that occurs after stroke is the main cause of brain damage. After cerebral ischemia, it causes energy production disorder in brain tissue, releases a large amount of excitatory neurotransmitters, and further produces oxidative free radicals and excitability. Toxicity, inflammatory factors and other effects lead to neuronal apoptosis or death, while neuroprotective drugs can prevent or slow these changes to reduce brain damage in patients (*Kurisu & Yenari, 2018*; *Saini, Guada & Yavagal, 2021*). In recent years, studies have consistently demonstrated a positive correlation between the PDE4D gene and the susceptibility to ischemic stroke (*Kim et al., 2009*; *Milton et al., 2011*; *Das, Roy & Munshi, 2016*; *Yasmeen et al., 2019*). The phosphodiesterase 4 inhibitor was identified to improve the synaptic plasticity of the neural circuit, regulate the metabolism of cells, prevent secondary brain damage caused by the recovery of blood flow in the ischemic area, and promote the functional recovery of injured neurons (*Zhang et al., 2014*; *Das, Roy & Munshi, 2016*; *Wang et al., 2018*). Therefore, phosphodiesterase 4 provides a novel therapeutic target for the treatment of stroke (*Sallustio et al., 2007*), and its comprehensive study will contribute to the development of therapeutic drugs for stroke (*Gretarsdottir et al., 2003*; *Sasaki et al., 2007*; *Xue et al., 2009*; *Soares et al., 2016*; *Chen et al., 2018*; *Yue et al., 2019*; *Ponsaerts et al., 2021*; *Vilhena et al., 2021*; *Xu et al., 2021*). The present article aims to provide an overview of the ongoing investigations on phosphodiesterase 4 as a promising pharmacological target for stroke treatment.

## SURVEY METHODOLOGY

This study conducted a literature search on phosphodiesterase 4 and stroke in Pubmed (https://pubmed.ncbi.nlm.nih.gov/) database and Google Scholar (https://scholar.google.com/), and then conducted an integrative scientometric review. The study focused on phosphodiesterase 4 as a potential therapeutic target for stroke.

The keywords comprised "stroke", "phosphodiesterase 4", "therapy", "cAMP", "brain injury" and "pathology", performed by crossing these descriptors using the Boolean operators "OR" and "AND". The inclusion criteria only considered white literature (articles and scientific notes) published in English between 1994 and 2024. Exclusion criteria include grey literature (monographs, dissertations, books, chapters, research published in conference proceedings) and review articles.

## Pathophysiology of stroke

The disease process of stroke involves the pathophysiological changes of many processes. Brain tissue ischemia and hypoxia lead to cell energy depletion and a reduction in ATP production, thereby triggering a cascade amplification effect due to an imbalance in the cAMP signaling pathway. Cytopathological changes caused by stroke include energy failure, cell ion homeostasis, acidosis, increased intracellular calcium content, increased free radical-mediated toxicity, excitotoxicity, increased production of arachidonic acid products, complement activation, destruction of the blood-brain barrier, activation of glial cells, leukocyte infiltration and apoptosis (*Macrez et al., 2011*; *Hankey, 2017*; *Khoshnam et al., 2017*; *Boursin et al., 2018*; *Wimmer, Zrzavy & Lassmann, 2018*) (Fig. 1). The interrelated and interacting effects of the above changes lead to ischemic necrosis in the brain and the emergence of severe ischemic areas, namely the ischemic central area and the ischemic penumbra. Within a few minutes of brain ischemia, the ischemic central area is exposed to the most severe ischemic brain tissue, resulting in irreversible necrosis of cellular energy depletion and apoptosis (*Radak et al., 2017*; *Sergeeva et al., 2018*; *Uzdensky, 2019*). This ischemic central area is surrounded by a less severe tissue area, blood flow is reduced and the corresponding function is lost, but the metabolic activity is still maintained. This area is called the ischemic penumbra (*Durukan & Tatlisumak, 2007*; *Catanese, Tarsia & Fisher, 2017*). The neuronal morphological changes in the ischemic penumbra are characterized by organelle expansion, collapse of the nucleus, and disruption of the plasma membrane, releasing the intracellular material into the extracellular space. The area of the ischemic penumbra can reach half of the initial total lesion area, and there are fewer severe ischemic areas. The focal ischemic infarction develops slowly, and the neurons may only undergo apoptosis for several hours or days rather than irreversible deaths in the central area of ischemia. It provides a chance for post-stroke treatment. In order to prevent further cell necrosis in the ischemic penumbra, it is possible to reduce post-stroke injury by rapidly restoring blood flow, reducing the proportion of apoptotic cells, repairing damaged neurons, and improving synaptic plasticity of neurons (*Vazquez-Garza et al., 2017*).

The pathways for activation of apoptosis include both intrinsic and extrinsic pathways, and past decades have provided a wealth of new information for the apoptotic process following ischemic stroke injury (*Unal-Cevik et al., 2004*; *Zhu et al., 2005*; *Broughton, Reutens & Sobey, 2009*). Glutamate excitotoxicity is an important factor in inducing ischemic injury, leading to an increase in intracellular calcium toxicity, while intracellular defense mechanisms cause activation of multiple cellular pathways to further induce apoptosis (*Tiwari et al., 2016*). For example, an energy-dependent cell pump fails due to a

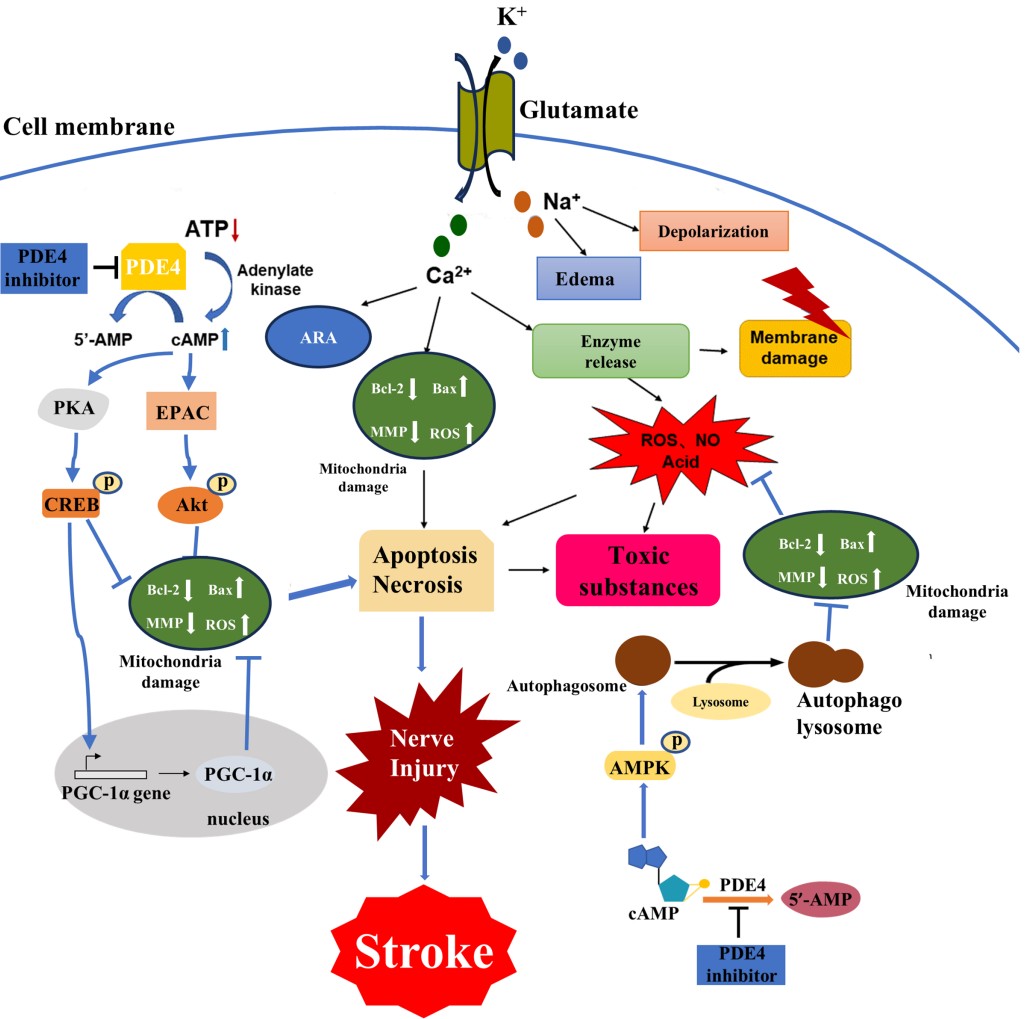

**Figure 1 Major cellular pathophysiological mechanisms of ischemic stroke injury.** Brain tissue ischemia and hypoxia lead to cell depletion, ATP production decreased, resulting in cAMP signal pathway disorder cascade amplification effect, a series of physiological and pathological changes, $Ca^{2+}$ and $Na^+$ flow to the cell, and $K^+$ outflow to the extracellular, $Na^+$ caused cells edema and depolarization, osmotic pressure change, $Ca^{2+}$ is caused by increased production of arachidonic acid, mitochondrial destruction leading to apoptosis death, enzyme release led to a large number of ROS, acidic substances also induced apoptosis, resulting in cell membrane rupture of a large number of toxic substances released to extracellular. PDE4 inhibitors could regulate the PKA/CREB pathway, the exchange protein activated by cAMP (EPAC)/Protein kinase B (Akt) pathway, the adenosine 5′-monophosphate (AMP) -activated protein kinase (AMPK) pathway and processes such as autophagy by inhibiting PDE4 to increase the cAMP level. Consequently, they could alleviate mitochondrial damage, reduce oxidative stress and ultimately decrease the loss of nerve cells.

decrease in the production of glucose-dependent ATP, causing flowing ionic substances to enter the cell causing cell swelling and further inducing an intrinsic apoptotic pathway (*Arundine & Tymianski, 2004*). There is increasing evidence that oxidative stress is closely related to the pathophysiological changes of post-stroke injury, and free radicals can react with DNA, proteins and lipids, leading to cell damage and dysfunction (*Narne, Pandey & Phanithi, 2017*). Processes such as lipid peroxidation (*Mattson, 2009*), inflammatory factor

production (*Kim et al., 2017*), leukocyte infiltration (*Gronberg et al., 2013*), have also been shown to be closely related to post-stroke injury, and their effects are related to apoptosis. The main purpose of neuroprotective drugs in the treatment of stroke is to reduce the damage of the ischemic penumbra and thus reduce the post-stroke injury. The effect is mainly through the cascade amplification effect of inhibiting the imbalance of cAMP signaling pathway, reducing apoptosis and repairing neuron synapses (*Venkat et al., 2018*).

The conclusion that the PDE4D gene is positively correlated with the risk of ischemic stroke is confirmed by more and more studies, and related factors such as organism genes and lifestyle may increase the risk of stroke (*Kumar et al., 2017*). Moreover, phosphodiesterase 4 inhibitors can improve the synaptic plasticity of the neural circuit, regulate the metabolism of cells, prevent secondary brain damage caused by the recovery of blood flow in the ischemic area, and promote the functional recovery of injured neurons for therapeutic purposes (*Hu et al., 2016*). Phosphodiesterase 4 provides a new drug target for the treatment of stroke. Therefore, in-depth study is helpful for the development of stroke treatment drugs.

## Current state of stroke treatment
### Surgical treatment of stroke
The narrow or even complete closure of the vertebral artery or internal carotid artery is the main cause of ischemic stroke. The clinical manifestations can be divided into three categories, namely complete stroke (CS), reversible ischemic neurological deficit (RIND), transient ischemic attack (TIA) (*Goyal et al., 2015*). Carotid endarterectomy (CEA) can remove carotid atherosclerotic plaque under direct vision to prevent arterial occlusion caused by plaque detachment (*Doig et al., 2015*). Since the 1980s, carotid angioplasty and stenting (CAS) has become a low-invasive treatment for atherosclerosis. Minimally invasive interventional measures (*Weimar et al., 2017*). Since 1967, Donaghy and Yasargil have successfully completed intracranial-external anastomosis in North America and Europe, respectively, making extracranial-cranial anastomosis a new type of surgical treatment for the prevention and treatment of cerebral ischemia (*Hornig, Dorndorf & Busse, 1986*). With current treatments, the risk of surgical treatment of stroke is extremely high, often at a risk that is greater than the benefit. Stroke surgical treatment can not improve the brain damage of stroke patients and the quality of life of patients, and postoperative complications are important reasons to limit their use.

### Medical treatment of stroke
In current clinical practice, the use of thrombolytic drugs is most common in the treatment of stroke patients. The use of thrombolytic drugs for thrombolytic therapy is one of the best treatments for the acute phase of ischemic stroke (*Rother, Ford & Thijs, 2013*). The use of anticoagulant drugs and fibrinolytic drugs are acute and early diagnosis and treatment of stroke (*Orbe et al., 2011*; *Kapil et al., 2017*). Currently, the first-line treatment for acute ischemic stroke is intravenous injection of recombinant tissue plasminogen activator (tPA) (*National Institute of Neurological Disorders and Stroke rt-PA Stroke Study Group, 1995*; *Jauch et al., 2013*; *Powers et al., 2018, 2019*). However, stroke patients who have reperfusion

of blood after ischemia in the brain can cause great brain damage. The use of antiplatelet drugs can relieve the hypercoagulable state of the blood, for the treatment of stroke patients (*Di Minno et al., 2013*). Thrombolytic-related drugs have no improvement on brain damage in stroke patients, and can not reduce secondary injury of patients. The patient's nerve function cannot be restored as well. Therefore, neuroprotective drugs that can alleviate brain damage in patients are becoming more and more research hotspots for treating stroke.

## PDE4 and stroke

### The structure, distribution and inhibitor development strategies of PDE4

Cyclic guanosine monophosphate (cGMP) and cyclic adenosine monophosphate (cAMP) are the major second messengers in cells, which play a key role in the life activities of cells such as cell metabolism, cell growth, memory, and neuronal signaling (*Beavo & Brunton, 2002*; *Martinez & Gil, 2014*; *Ricciarelli & Fedele, 2018*; *Cardarelli et al., 2019*). There are strict regulatory pathways for the synthesis and degradation of cGMP and cAMP in cells. Phosphodiesterase (PDEs) are the only pathways for the degradation of cGMP and cAMP in cells. They can participate in the cGMP and cAMP hydrolysis and open loop, corresponding to the production of no active form 5′-GMP and 5′-AMP (*Tello-Montoliu et al., 2012*). PDEs belong to the super-enzyme family, and 21 genes encode (*Agarwala et al., 2016*). PDEs are classified according to their enzymatic kinetic properties, different amino acid sequences, sensitivity to inhibitors, intracellular localization and tissue expression patterns, and substrate specificity. PDEs have more than 50 different types of PDE isozymes. Specific PDE isozymes can regulate specific cellular functions. Different subtypes in the same family can be distributed and expressed in different tissue cells, precisely regulating cell function (*Murthy & Mangot, 2015*; *Agarwala et al., 2016*). At present, the research on PDE4 is the most extensive and in-depth. The research confirms that it is concentrated in the cortex, hippocampus and striatum in the central nervous system of the body, and is highly specific to cAMP (*Agarwala et al., 2016*; *Klussmann, 2016*).

The PDE4 family consists of four subtypes, corresponding to PDE4A, PDE4B, PDE4C and PDE4D, which can be divided into three types: long, short, super-short and dead-short. The super-short form of PDE4 has no upstream conserved domains 1 (UCR1) and upstream conserved domains 2 (UCR2), a highly conserved domain that degrades cAMP, while short form PDE4 lacks UCR1 and long form PDE4 carries a highly conserved domain (UCR1 and UCR2) (*Gurney, D'Amato & Burgin, 2015*). Part of the catalytic domain is missing from the dead-short form. There is a linker region 1 (LR1) between UCR1 and UCR2, and UCR2 is connected to the catalytic zone through linker region 2 (LR2) (*Gavalda & Roberts, 2013*). The UCR1 is composed of approximately 60 amino acids, and the UCR2 is composed of around 80 amino acids. The UCR1 and UCR2 may interact with each other through electrostatic forces. The phosphorylation of serine on the UCR1 by protein kinase A (PKA) can regulate this electrostatic force. The phosphorylation by PKA will activate the interaction between UCR1 and UCR2. Depending on the different splicing isoforms, the hydrolysis of cAMP can be increased by two to four times. Moreover,

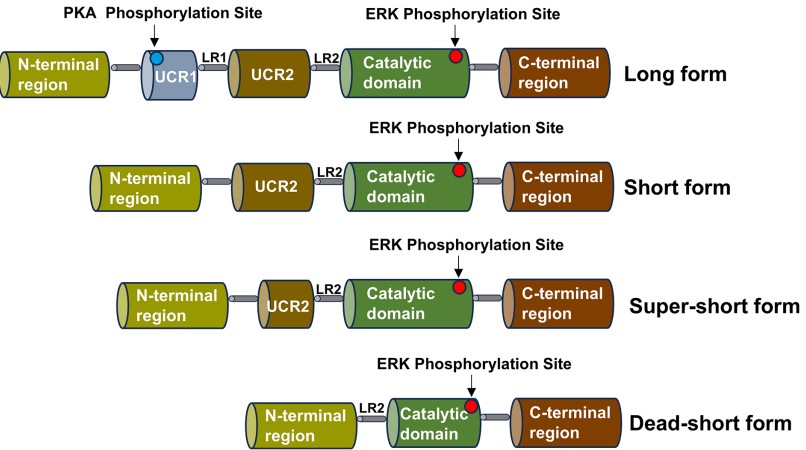

**Figure 2 Schematic diagram of the structure of the PDE4 family enzymes.** The structure shared by the four subtypes of the PDE4 family is displayed, including the UCR1 and UCR2 domains, the LR1 and LR2 regions, the catalytic domain, the N-terminal region and the C-terminal region. Various isoform forms generated by the use of different promoters or alternative splicing are presented, and these isoforms can give rise to the long, short, super-short and dead-short forms of these enzymes.

the active site on the UCR2 can control the entry of cAMP. Small molecules can bind to the active sites on the PDE4 and interact with the specific amino acids of the UCR2. PDE4 is highly expressed in nerve cells and inflammatory cells, such as lymphocytes, macrophages, and mast cells, and is enriched in the nucleus, cell scaffold, and plasma membrane of cells (*Teixeira et al., 1997*). The schematic diagram of the structure of the PDE4 family enzymes is shown in Fig. 2.

The catalytic region of PDE4 is composed of approximately 380 amino acid residues, which fold into 17 α-helices (H0-H16) and form three closely bound domains. Among them, H1-H7 constitute the first domain, H8-H11 form the second domain, and H12-H16 make up the third domain. The substrate binding site of PDE4 is located at the junction of these three domains and forms three pockets, namely the M, Q, and S pockets. Research shows that the pockets in the catalytic region of PDE4 contain three sub-regions. There is a hydrophobic region containing two amino residues, Gln443 and Asn395, and these two amino residues play an important role in the specific recognition of nucleotides. The metal region is formed by two positive ions ($Zn^{2+}$, $Mg^{2+}$) connecting polar residues and coordinated with 6 water molecules. The solvent region is mainly formed by hydrophilic amino acids and a network filled with water molecules. Therefore, according to the properties of the pockets in the catalytic region of PDE4, the structure of PDE4 inhibitors needs to possess two characteristics. The structure of PDE4 inhibitors requires a planar ring to interact with the phenylalanine in the hydrophobic pocket of PDE4 and form a hydrogen bond with the amino group in the glutamine in the active pocket of PDE4 (*Card et al., 2004*).

The N-terminal region of PDE4 isoenzymes regulates their intracellular localization. The sequence of this region determines the interaction of each isoenzyme with other proteins and its anchoring to specific subcellular compartments, including its binding to

A-kinase anchoring proteins (AKAPs). The above-mentioned functions enable the existence of macromolecular signaling complexes, thereby allowing for the specific regulation of pathophysiological processes that are relevant to therapy. For example, the N-terminal region enables PDE4A4 and PDE4D4 to specifically interact with the SRC family of tyrosine kinases, allows PDE4A4 to bind to hepatitis B virus-associated protein 2, enables PDE4D5 to interact with β-arrestin, facilitates the association of PDE4D3 with both muscle-specific A-kinase anchoring protein (mAKAP) and AKAP450, and also makes it possible for PDE4D5 to interact with the receptor for activated C kinase 1 (RACK1). The precise localization of PDE4 enzymes is crucial for the interaction between cAMP and specific proteins. These specific proteins can induce conformational changes, alter catalytic activity, and even influence the affinity of inhibitors. All these mechanisms allow for the dynamic regulation of the quantity, localization, and activity of PDE4 enzymes so as to adapt to and respond to the spatio-temporal modulation of cAMP signaling.

The catalytic domain of PDE4 is connected to control region 3 (CR3) and then linked to the C-terminal region. There are nine subtypes of PDE4D, namely PDE4D1-9. It is similar in structure to PDE4B and also has long-chain, short-chain, and ultrashort-chain isoenzymes. Due to the high homology among PDE4 subtypes and the strict conservation of the amino acid sequences among the active sites of these subtypes, it has become rather difficult to discover inhibitors targeting specific subtypes. Therefore, by studying the structures of the N-terminal and C-terminal regulatory domains, especially the influence of UCR2 and CR3 on the structure of PDE4B inhibitors, highly selective PDE4B inhibitor molecules can be designed. The way in which small molecule inhibitors of PDE4 bind to CR3 may be of great importance for the selectivity among PDE4 subtypes (PDE4B and PDE4D). The selectivity of PDE4B can be achieved by regulating CR3 at the C-terminal active site, which is determined by the polymorphism of individual amino acids in CR3. Structural studies have shown that it has flexibility and possesses multiple orientation sites in the closed conformation. *Lorimer et al. (2009)* found in their research that CR3 exists in the protein constructs of the crystallized PDE4 catalytic domain, yet it is generally disordered within the lattice, and they hypothesized that it might be related to the weak interaction between the helix itself and the catalytic domain. Meanwhile, they also discovered that small molecule ligands can interact with different amino acid residues on the CR3 helix (*Lorimer et al., 2009*). These results indicate that CR3 can regulate the biological activity of PDE4 isoenzymes, providing a direction for the drug design of selective PDE4B inhibitors.

Most of the existing selective PDE4 inhibitors can selectively inhibit PDE4 among PDE1 to PDE11. However, their selectivity among PDE4 subtypes (PDE4A to PDE4D) is not very good. Therefore, some people have speculated whether a certain subtype of PDE4 is responsible for causing vomiting. Some studies have found that mice with a deletion of the PDE4D gene can shorten the duration of anesthesia, while mice with a deletion of the PDE4B gene cannot. Hence, it is hypothesized that the inhibition of the PDE4D subtype may lead to vomiting (*McCluskie et al., 2006*). In 2009, it was reported that a 2-aryl pyrimidine derivative had an inhibitory activity against PDE4B that was more than 100

times higher than its inhibitory activities against the other three subtypes. Nevertheless, it still caused vomiting in animal experiments. Therefore, we cannot simply attribute the side effects of PDE4 inhibitors to their poor subtype selectivity, and the specific reasons remain to be further explored (*Naganuma et al., 2009*).

To obtain PDE4 inhibitors with greater development potential based on the characteristics of the PDE4 active cavity. When modifying the structure of FCPR03, it is considered to replace the nitrogen atom of the amide bond with a carbon atom to increase the lipophilicity of the molecule, and replace the cyclopropylmethoxy group with a secondary amine, tertiary amine or heterocyclic ring, which increases the inhibitory activity against PDE4B. With the aim of obtaining PDE4 inhibitor analogues that exhibit higher bioavailability, researchers have engaged in the design and synthesis of a new series of arylbenzylamine derivatives. The arylbenzylamine derivatives featuring a pyridin-3-amine side chain display favorable inhibitory activities against human PDE4B1 and PDE4D7 subtypes. Additionally, the interaction between the inhibitors and the UCR2 of PDE4B1 not only enables partial inhibition of PDE4 but also leads to a remarkable enhancement in oral bioavailability (*Naganuma et al., 2009*). Replacing the cyclopropylmethoxy group of the lead compound FCPR03 with cyclopropylmethylamine resulted in higher bioavailability and good blood-brain barrier permeability (*Xia et al., 2022*). The molecular docking approach was employed to investigate the interaction between pyrimidine diaryl-substituted PDE4B inhibitors and the enzyme. It was observed that the introduction of a five-membered cyclic hydrocarbon moiety substituted with a larger electronegative group at the 2-position of the pyrimidine ring within this series of compounds could augment the inhibitory activity. Moreover, the incorporation of an aryl moiety substituted with a hydrogen bond donor or a hydrogen bond acceptor at the 6-position of the pyrimidine ring could also enhance the compound's activity. Researchers designed and synthesized a series of indole heteroaryl compounds based on the docking mode of classical compound molecules (roflumilast and apremilast) with PDE4B. It was discovered that the introduction of a methoxy group on the indole could enhance the activity of PDE4 inhibitors (*Sunke et al., 2019*).

PDE4 is abundant in the striatum, hippocampus, cortex and cerebellum. PDE4A, PDE4B and PDE4D are common subtypes in the brain, while PDE4C exists only in certain regions of the brain and is also the subtype with the lowest distribution of PDE4 in the brain (*Lakics, Karran & Boess, 2010*). The roles of cAMP and PDE4 in the increased permeability of endothelial cells in the blood-brain barrier after injury have been fully demonstrated. The research findings indicate that PDE4 can mediate cell adhesion molecules (CAMs) in peripheral cells, which may provide information for further studies on the mechanisms of endothelial cells and neurons in the blood-brain barrier and bring new directions for the application of PDE4 inhibitors in the prevention and treatment of stroke (*Rampersad et al., 2010*; *Blokland et al., 2019*).

### Overview and optimization strategies of PDE4 inhibitors

In the 1980s, academic circles and pharmaceutical companies began to devote themselves to the research and development of PDE4 inhibitors. In the 1990s, the first-generation

PDE4 inhibitor rolipram was developed. However, it had serious gastrointestinal side effects in clinical use, such as nausea, vomiting, and diarrhea, which led to severe restrictions on its clinical application. In the 1990s, Schering AG of Germany developed the first-generation PDE4 inhibitor rolipram, and subsequently, Meiji Seika of Japan also developed similar drugs (*Dal Piaz & Giovannoni, 2000*). But these drugs had serious gastrointestinal side effects in clinical use, such as nausea, vomiting, and diarrhea, which restricted their clinical application significantly. In the 2010s, the second-generation PDE4 inhibitors were successively approved for marketing, such as roflumilast, apremilast, and crisaborole (*Abdulrahim et al., 2015*; *Crocetti et al., 2022*). These drugs have shown certain efficacy in treating diseases such as chronic obstructive pulmonary disease (COPD), psoriatic arthritis, and atopic dermatitis (*Li, Zuo & Tang, 2018*). However, they still have some limitations. For example, although roflumilast can improve the symptoms of COPD patients to a certain extent, it can cause adverse reactions such as weight loss and sleep disorders, which affect the patients' tolerance and long-term treatment compliance. In the head-to-head clinical trials with similar drugs, the efficacy and safety of crisaborole have not been proven to be superior (*Crocetti et al., 2022*). In recent years, researchers have been continuously exploring new drug design strategies and administration routes to further improve the efficacy and safety of PDE4 inhibitors. As a result, inhalable PDE4 inhibitors CHF6001 have emerged.

Different subtypes of PDE4 have distinct expression patterns in specific tissues, cell types, and intracellular localizations. Therefore, the development of PDE4 inhibitors targeting specific subtypes can help reduce adverse reactions by selectively inhibiting the relevant subtypes. Commonly used non-specific PDE4 inhibitors exhibit strong binding affinity to the high-affinity site of the enzyme (HPDE4), thereby leading to an increase in cAMP levels, and the elevation of cAMP levels is an important cause of adverse reactions (*Jeon et al., 2005*). Tissue-selective modulation can improve the treatment response and effectiveness by directly applying drugs to the affected brain regions. Progressive neuronal loss over time will reduce the drug response. By focusing on specific brain regions related to PD pathology and reducing off-target effects in other parts of the body, the tissue-selective modulation of PDE4 inhibitors aims to minimize these adverse side effects (*Chen et al., 2018*). Compared with rolipram, many newly developed novel PDE4 inhibitors have high affinity for specific subtypes of PDE4 and have mitigated the emetic effect. The inhibitory effect of FCPR16 on PDE4 does not cause vomiting in beagle dogs at a dose of 3 mg/kg, indicating that FCPR16 has the potential to reduce the emetic effect (*Zhou et al., 2016*). FCPR03 has been structurally optimized to generate two new aminophenyl ketone-structured compounds (9C and 9H). They possess high potency, activity comparable to rolipram, higher bioavailability, low thrombogenicity, and improved blood-brain barrier permeability, and can also improve memory and cognitive dysfunction in mouse models of Alzheimer's disease (*Zou et al., 2017*). Oral administration is the most common route for delivering PDE4 inhibitors, as it offers the advantages of convenience, patient preference, cost-effectiveness, and ease of large-scale production of oral dosage forms. Oral PDE4 inhibitors need to survive in the harsh gastrointestinal environment, penetrate the intestinal epithelium, and bypass liver metabolism (including

the first-pass effect) before reaching the systemic circulation. One of the main reasons why there are no specific and effective treatments for neurodegenerative diseases is that drugs cannot pass through the blood-brain barrier (*Correia et al., 2022*). Apremilast has been approved for the treatment of moderate to severe plaque psoriasis and psoriatic arthritis. However, due to the inhibitory effect of PDE4 in non-target tissues, dose titration is required to avoid gastrointestinal side effects such as nausea and diarrhea (*Abdulrahim et al., 2015*).

Currently, numerous PDE4 inhibitors are still under clinical investigation. For example, the compound crisaborole, which has better transdermal efficacy. Despite its moderate potency in inhibiting enzyme activity (IC50 = 490 nM), it has demonstrated excellent anti-inflammatory activity both *in vitro* and *in vivo* (*Akama et al., 2009*). Studies on AD-301 (NCT No. NCT02118766) and AD-302 (NCT No. NCT02118792) have indicated that twice-daily treatment with 2% ointment is most effective in alleviating symptoms (*Luger et al., 2022*). Unlike systemic treatment, topical treatment with crisaborole does not cause gastrointestinal adverse reactions. This may be because the molecule is rapidly metabolized into two inactive compounds after topical application (*Zane et al., 2016*). Crisaborole is being evaluated in clinical trials for its efficacy and safety in treating psoriasis with different concentrations of topical preparations in patients with psoriasis vulgaris. Ensifentrine is a novel, selective, dual inhibitor capable of inhibiting both PDE3 and PDE4. Clinical studies (NCT No. NCT04535986 or NCT04542057) have indicated that it is well tolerated and is not associated with the increase in gastrointestinal tolerance issues related to oral PDE4 treatment. Ensifentrine, with its novel inhibitory mechanisms of PDE3 and PDE4, will be a valuable addition to the limited existing treatment mechanisms for patients with COPD. In the two trials, Ensifentrine significantly improved lung function and reduced the exacerbation rate among a broad range of COPD patients (*Anzueto et al., 2023*). Cilomilast may play a very good role in the treatment of COPD, yet its clinical trials have been suspended for many years. After the commercialization of roflumilast, cilomilast quickly entered the clinical stage for the treatment of COPD and asthma. Cilomilast is regarded as the most promising candidate drug, although the results are not sufficient to permit its marketing (*Martina, Ismail & Vesta, 2006*). Ibudilast has now completed a Phase II clinical trial in multiple sclerosis, and the results show that it appears to have a therapeutic effect on brain atrophy, possibly slowing disease progression. However, ibudilast has also shown side effects common to PDE4 inhibitors in clinical studies, such as gastrointestinal symptoms and headaches (*Fox et al., 2018*). Two promising compounds, MK0952 and BPN14770, and the drug Roflumilast, have been developed for the treatment of Alzheimer's disease and are in Phase 2 and Phase 1 clinical trials, respectively. BPN14770, in particular, has been involved in several clinical trials, the last of which (NCT03817684) is still in phase 2 activity studies (*Gallant et al., 2010*; *Devadiga & Bharate, 2022*). GSK356278 completed two Phase I clinical studies (one alone and one in combination with Rolipram) in 2017 for the treatment of Huntington's disease (HD), an autosomal progressive neurodegenerative disorder (*Wyant, Ridder & Dayalu, 2017*). The structure of PDE4 inhibitors are shown in Fig. 3.

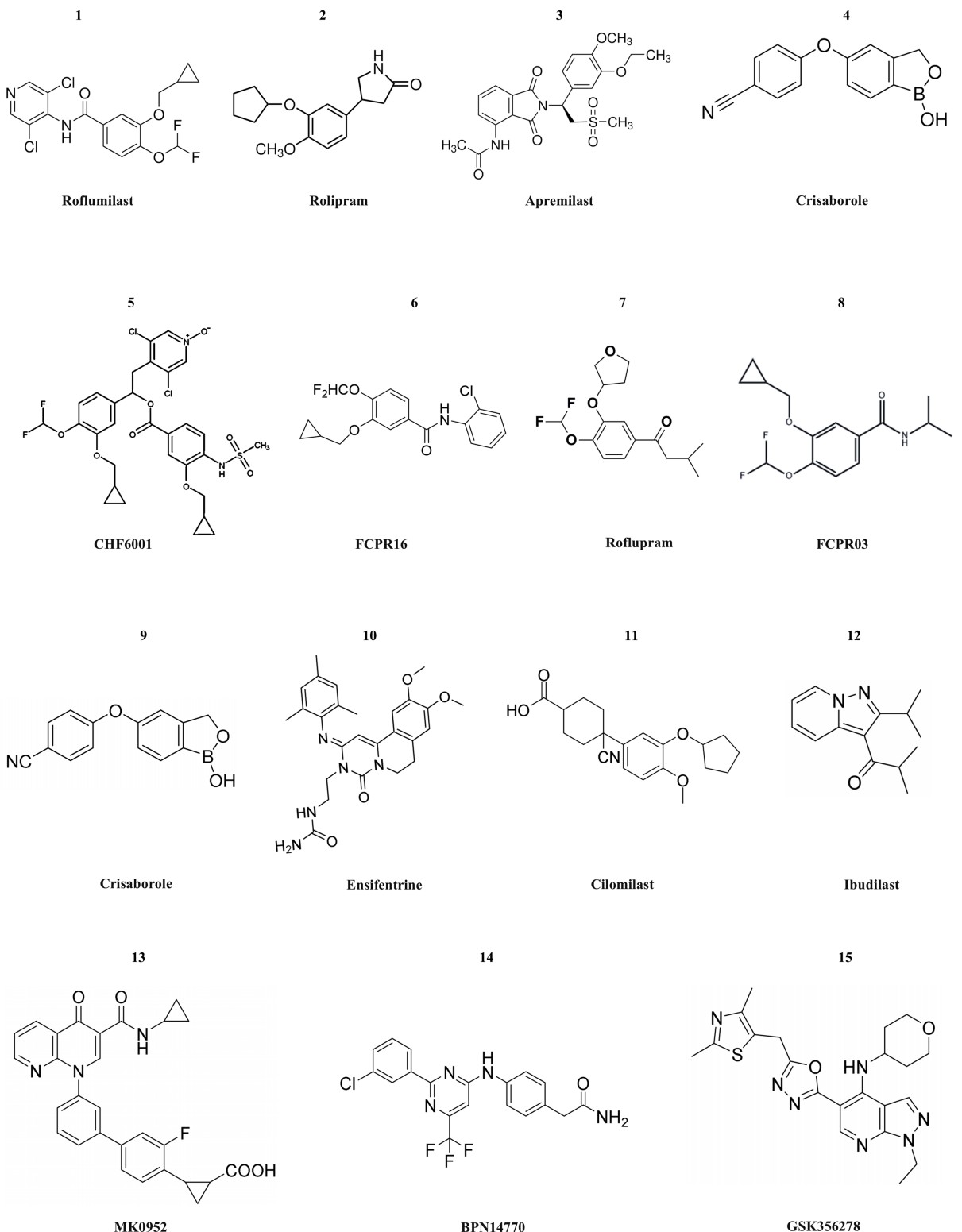

**Figure 3 The structural formula of PDE4 inhibitors.** The structural formulas of the PDE4 inhibitors mentioned in the text are as follows, including roflumilast, rolipram, apremilast, crisaborole, CHF6001, FCPR16, roflupram, FCPR03, crisaborole, ensifentrine, cilomilast, ibudilast, MK0952, BPN14770 and GSK356278.

There are still numerous problems to be solved in the process of developing PDE4 inhibitors into drugs. For instance, gastrointestinal adverse reactions are the most common side effects in clinical practice. This is because PDE4 is also expressed in gastrointestinal tissues, and the inhibitory effect of drugs on gastrointestinal PDE4 may lead to gastrointestinal dysfunction. The issue of safety also needs to be taken into consideration by researchers. PDE4 inhibitors may pose potential risks to the cardiovascular system as they might affect the function of vascular endothelial cells or the signal transduction of cardiomyocytes. There are significant differences in the pathophysiological processes among different diseases such as COPD, psoriatic arthritis, and atopic dermatitis. Genetic differences among patients can also influence the efficacy of PDE4 inhibitors. The PDE4 gene in different individuals has polymorphisms, which may result in different binding affinities between drugs and targets or differences in the drug metabolism process. In clinical treatment, patients may use multiple anti-inflammatory drugs simultaneously. When PDE4 inhibitors are used in combination with other anti-inflammatory drugs such as glucocorticoids and non-steroidal anti-inflammatory drugs, synergistic or antagonistic effects may occur. Since many diseases (such as COPD, psoriatic arthritis, *etc.*) require long-term drug treatment, long-term safety monitoring of PDE4 inhibitors is a difficult point. Long-term use may cause some potential adverse reactions to gradually emerge, such as liver and kidney function impairment, abnormal immune system, *etc*. During long-term treatment, patients may develop tolerance to the efficacy of drugs. As time passes, the therapeutic effect of drugs may gradually weaken, which may be related to various factors such as the body's adaptive changes and disease progression. For example, in the treatment of atopic dermatitis, after using PDE4 inhibitors for a period of time, patients may experience recurrence of the disease or the treatment effect may not be as obvious as it was in the initial stage (*Devadiga & Bharate, 2022*).

Researchers have already taken a lot of measures to improve the problems related to PDE4 inhibitors. In response to gastrointestinal adverse reactions, the pharmaceutical formulation can be optimized by designing sustained-release or targeted formulations of PDE4 inhibitors. Some drugs for protecting the gastrointestinal tract can be used in combination with PDE4 inhibitors. Precision medicine can be carried out based on different subtypes of diseases and the genetic characteristics of patients. PDE4 inhibitors targeting the pathophysiological processes of different diseases can be developed. For example, in light of the characteristics of joint inflammation and bone destruction in psoriatic arthritis, PDE4 inhibitors that are capable of simultaneously regulating the inflammatory response and bone metabolism can be developed. Alternatively, for the immune abnormalities and impaired skin barrier function in atopic dermatitis, drugs that can improve the skin's immune microenvironment and repair the skin barrier can be designed.

Newly developed drug delivery technologies can overcome the above-mentioned drawbacks of oral PDE4 inhibitors. These technologies include liposomes, nanonization, solid lipid nanoparticles, self-emulsifying drug delivery systems, and self-microemulsifying drug delivery systems. The research and development of PDE4 inhibitors can obtain the key amino acid residues for enhancing efficacy and selectivity by analyzing the co-crystal

complexes of compounds and proteins. Adverse reactions associated with PDE4 inhibitors can also be reduced through strategies such as partial PDE4 inhibition. Compared with potent inhibitors, partial PDE4 inhibitors can provide a safer pharmacological profile by having different affinities for different PDE4 subtypes (*Larsen et al., 2020*). PDE4 inhibitors need to have better blood-brain barrier permeability and the ability to achieve brain-specific delivery to reduce adverse reactions related to peripheral PDE4 inhibition. For example, roflumilast has been proven to improve cognitive abilities and provide anti-neural damage capabilities. Although small molecule PDE4 inhibitors have been extensively studied as treatments for several human inflammatory diseases, only a few have been able to obtain approval due to the side effects resulting from the widespread expression of PDE4 in many tissues. Therefore, an issue that remains to be addressed is achieving tissue and cell specificity for specific therapeutic targets. Strategies for improving PDE4 inhibitors include nanomolar activity at the PDE4 catalytic site, low affinity for high affinity rolipram binding state (HARBS), selectivity for PDE subtypes, potent tumor necrosis factor (TNF-$\alpha$) inhibitory activity, low brain penetration, combination therapy, alteration of the mode of administration.

### Polymorphism of PDE4D gene

Stroke is a disease influenced by numerous factors such as age, hypertension, diabetes, *etc*. Studies have revealed that it is closely associated with genetic factors (*Zhai et al., 2017*). The PDE4D gene is located at 5q12 and has a length of 1.5 Mb. It can encode at least nine isomers and contains no less than 22 exons (*Munshi & Kaul, 2008*). PDE4D can specifically hydrolyze cAMP to attenuate the cell signaling activity of G protein-coupled receptors, and elevated cAMP levels *in vivo* can activate protein kinase A, which in turn activates protein phosphorylase, regulates intracellular and extracellular $Ca^{2+}$ levels, and inhibits the production of inflammatory factors and platelet activation, while reducing the degree of atherosclerosis (*Kraft et al., 2013*). In 2002, *Gretarsdottir et al. (2003)* discovered that the strk1 locus on chromosome 5q12, which represents the PDE4D gene, was associated with ischemic stroke by genome-wide scanning. The study of single nucleotide polymorphism (SNP) found that multiple loci of PDE4D were associated with stroke, including SNP41, SNP45, SNP56, SNP83, SNP87 and SNP89, and all SNPs and SNPs. The markers are located in the intron region at the 5′ end of the PDE4 gene (*Li et al., 2010*; *Wang et al., 2017*). Compared with stroke patients, the total mRNA expression of PDE4D1, PDE4D2, and PDE4D5 in B lymphocytes in normal population was reduced (*Gretarsdottir et al., 2003*).

*Grond-Ginsbach et al. (2008)* used microarray to study the transcription level of PDE4D in blood mononuclear cells in stroke patients. It was reported that the total level of PDE4D transcription in stroke patients was lower than that in the normal population. But its research did not use internationally recognized real-time quantitative PCR or detection of protein expression levels, so the results are controversial (*Grond-Ginsbach et al., 2008*). After the results of *Gretarsdottir et al. (2003)* were published, research groups around the world could not form a consensus in repeated experiments or related research. A study by *Woo et al. (2006)* reported a racial difference between PDE4D and cardiogenic stroke, and

genetic polymorphisms in rs2910829 revealed that it was associated with stroke in black and white studies, and rs152312 only reveals related to cardiogenic stroke in white races. A study conducted by *Staton et al. (2006)* in Australia found that PDE4D was strongly associated with stroke in SNP83, SNP89, and SNP87, while SNP41, SNP56, and SNP45 did not show significant correlation. In addition, meta-analysis of *Nath et al. (2023)* suggest that SNP45, SNP83, and SNP89 polymorphisms may increase stroke susceptibility in Asians, but not in Caucasian populations. The genotyping of SNP 45,83,89 polymorphism can be used as a predictor of the occurrence of stroke (*Nath et al., 2023*). Inhibiting PDE4D in stroke protects the blood-brain barrier, reducing inflammation and thrombosis. Inflammation plays an important role in the occurrence and development of atherosclerosis, and PDE4 is significantly expressed in inflammatory cells during cerebral ischemia. Variations and polymorphisms in the PDE4D gene have been studied in different stroke populations to identify any associated risks. The above literature shows that the changes of PDE4D are associated with stroke, and the direct involvement of PDE4D is strongly supported by association and expression analyses. The association analyses (single marker and haplotype analyses) proposed by researchers support the view that PDE4D increases the risk of stroke. Additionally, significant dysregulation of multiple PDE4D subtypes has been observed in the affected individuals. Some researchers have put forward that PDE4D participates in the pathogenesis of stroke through atherosclerosis. PDE4D is expressed in the important cell types in atherosclerosis and regulates the second messengers that play significant roles in the pathogenesis of atherosclerosis. Perhaps by inhibiting PDE4D with small molecule drugs or inhibiting one or more PDE4D subtypes, the risk of stroke in the population susceptible to the PDE4D genotype could be reduced. It may be possible to intervene in advance or take preventive measures by detecting the levels of risk genes to reduce the occurrence of stroke. The relationship between PDE4D gene polymorphism and ischemic stroke remains to be further studied, and the existing research is summarized in Table 1.

### Relationship between PDE4 and inflammatory response in stroke injury

After ischemic stroke injury, some cell groups that are fixed in the tissue secrete pro-inflammatory mediators, including glial cells, neuronal cells, and endothelial cells (*Sumin et al., 2017*). Activation of transcription factors leads to increased cytokines, elevated protein levels, and increased endothelial CAMs in brain tissue after stroke (*Li et al., 2017a*). The occurrence of inflammation after stroke is mainly due to the role of microglia, especially in the central zone of ischemia and the ischemic penumbra junction (*Huang, Upadhyay & Tamargo, 2006*). Activated microglia produce a large number of pro-inflammatory factors as toxic metabolites and enzymes to induce a series of cellular responses, and astrocytes are also involved (*Karthikeyan et al., 2016*). The proinflammatory cytokines IL-8, IL-6, IL-1 and TNF-α cause leukocytes to accumulate in the ischemic central zone and ischemic penumbra during ischemic stroke reperfusion. In turn, the chemotaxis of neutrophils is activated, aggravating cerebral ischemia and generating a large number of oxidative metabolites to produce neurotoxicity, further leading to brain damage (*Jin et al., 2013*; *Zarruk, Greenhalgh & David, 2018*).

**Table 1 PDE4D gene association with stroke.**

| Researchers | Study population | Stroke type |
|---|---|---|
| *Gretarsdottir et al. (2002)* | Iceland | Large artery atherosclerosis, cardioembolic |
| *Staton et al. (2006)* | Australian | All ischemic stroke |
| *Woo et al. (2006)* | Caucasian & Afro-American | All types of stroke |
| *Song et al. (2006)* | US females (<50 years) | All ischemic stroke, cardioembolic.atherosclerotic |
| *Kostulas et al. (2007)* | Sweden | All types of stroke |
| *Milton et al. (2011)* | Australian | Cardioembolic |
| *Xu et al. (2010)* | Asian | All types of stroke |
| *Kuhlenbäumer et al. (2006)* | Gemany | All ischemic stroke. Large artery atherosclerosis, cardioembolic |
| *Lõhmussaar et al. (2005)* | Germany | All ischemic stroke, Large artery atherosclerosis, cardioembolic |
| *Yoon et al. (2011)* | Korean | All ischemic stroke |
| *Wang et al. (2017)* | Han Chinese | Atherosclerosis stroke |
| *He et al. (2013)* | Chinese | Stroke |
| *Shao et al. (2015)* | South Eastern Han Chinese | Stroke |
| *Yan et al. (2014)* | Asian and Caucasian | Stroke |

Studies have shown that reducing the expression of pro-inflammatory factors while inhibiting inflammatory response is a potential therapeutic strategy for cerebral ischemia-reperfusion (*Lambertsen, Biber & Finsen, 2012*; *Santos Samary et al., 2016*). PDE4 is highly expressed in nerve cells and inflammatory cells such as lymphocytes, macrophages, and mast cells. PDE4 inhibitors have been widely recognized for their anti-inflammatory effects. PDE4 inhibitors inhibit the production of various types of cytokines *in vivo*, mainly related to the increase of cAMP levels *in vivo*. Although PDE4 inhibitors can reduce serum TNF-α levels, the underlying mechanism remains unclear (*Zou et al., 2017*). Studies have shown that inhibition of endogenous TNF activity can reduce cerebral infarct size and reduce cerebral ischemia-reperfusion injury (*Esposito & Cuzzocrea, 2009*; *Sumbria, Boado & Pardridge, 2013*).

The PDE4 inhibitor roflumilast exerts a down-regulation of neutrophil adhesion by down-regulating the expression of neutrophil β2-integrin, impeding the interaction between endothelial cells and neutrophils (*Hoymann et al., 2009*). Roflumilast can reduce the release of related inflammatory mediators by inhibiting the activation of the transcription factor NF-κB, such as IL-8, IL-6, IL-1 and IL-1β (*Sanz et al., 2007*). PDE4 inhibitor Rolipram can counteract cerebral ischemic injury by reducing neutrophil invasion and reducing the expression of pro-inflammatory cytokines IL-1β and TNFα (*Kraft et al., 2013*). In conclusion, PDE4 inhibitors can protect the brain tissue of stroke patients by reducing the inflammatory response of the ischemic penumbra in stroke injury, thereby achieving therapeutic goals.

### PDE4 and cascade amplification reaction of cAMP signal

When the ischemic penumbra after cerebral ischemia in stroke patients restores blood perfusion for a period of time, it will aggravate brain tissue damage and accelerate disease

progression. After cerebral ischemia, the brain tissue is affected by hypoxia, resulting in impaired cellular energy production, with insufficient energy production and accelerated consumption as well as insufficient production of adenosine triphosphate, thereby leading to a cascading amplification of cAMP signaling pathway imbalance. This further induces a series of metabolic changes in cell physiology and biochemistry, such as an abrupt reduction of aerobic metabolism, a massive increase of oxygen free radicals, serious cell edema and other pathological changes (*Zhang et al., 2012*).

PDE4 inhibitors can specifically block the hydrolysis of cAMP by acting on intracellular PDE4, increasing its concentration in cells. Activation of PKA results in the entry of the C subunit into the nucleus, further catalyzing the phosphorylation of the serine residue of the cAMP-response element binding protein (CREB). Then the cAMP response element on the DNA target gene binds to phosphorylated CREB, ultimately regulating the initiation of gene transcription (*Ding et al., 2012*). PDE4 inhibitors can increase the level of cAMP *in vivo*, thereby blocking the cascade amplification effect of cAMP signaling pathway, preventing a series of pathological metabolic changes of cells, reducing cellular oxygen free radicals, and reducing cell edema to achieve the purpose of treating post-stroke injuries (*Vincent et al., 2012*). Roflumilast, a PDE4 inhibitor, can improve the cognition of patients with acquired brain injury (ABI) by regulating the activity in the cAMP-Phosphokinase A-Ras-related C3 botulinum toxin substrate (RAC1) inflammation pathway. Moreover, PDE4 inhibitors can also directly enhance network plasticity and alleviate degenerative processes and cognitive dysfunction by increasing the activity of the canonical cAMP/phosphokinase-A/cAMP responsive element binding protein (cAMP/PKA/CREB) plasticity pathway (*Schreiber, Hollands & Blokland, 2020*).

### PDE4 and neurotrophic factors in stroke injury

Neurotrophic factors are a class of protein molecules that play an important role in the development, survival and apoptosis of neurons. Currently, there are two main types, namely brain-derived neurotrophic factor (BDNF) and nerve growth factor (NGF). And BDNF is highly expressed in the central nervous system and is enriched in the hippocampus, hypothalamus, substantia nigra, striatum, cerebellum, cortex, *etc.* (*Kowianski et al., 2018*). After brain injury, the neurons in the ischemic central zone and the ischemic penumbra change the permeability of the plasma membrane due to hypoxia-ischemia, and increased extracellular $Ca^{2+}$ influx produces a large number of excitatory amino acids, further leading to the initiation of cellular response mechanisms and up-regulation of BDNF and its receptor expression (*Zhao et al., 2017*). Ferrer showed that the expression of TrkB in astrocytes around the infarcted tissue increased after stroke, and the BDNF immune response also increased transiently, indicating that BDNF is involved in the protection process of the body after stroke injury (*Ferrer et al., 2001*). The expression of TrkB and BDNF in the brain tissue of patients with stroke increased, which can be confirmed to protect neurons against ischemia and hypoxia (*Ferrer et al., 2001*). *Han et al. (2000)* found that by continuously injecting BDNF into the lateral ventricle, the total area of cerebral infarction in the animal model of stroke was reduced by 33%, and the area of cortical infarction was reduced by 37%.

Kiprianova found that injection of BDNF into the lateral ventricle immediately after cerebral ischemia prevented apoptosis in neurons in the hippocampal CA1 region of mice (*Nitta et al., 1999*). These findings suggest that the mechanism by which BDNF reduces stroke injury may be to stabilize intracellular $Ca^{2+}$ concentration by regulating the expression of calcium-binding protein and to deliver nutrients to injured neurons to increase survival. Current studies indicate that cAMP activates PKA to release its catalytic subunit and phosphorylates CREB at the ser133 site to produce its activated form, p-CREB (*Naqvi, Martin & Arthur, 2014*). The importance of CREB is reflected in its damage in brain tissue leading to degeneration of neurons. cAMP response element (CRE) promotes the release of neuronal survival factors BDNF and bcl-2 (*Lee, 2015*). PDE4 inhibitors can directly activate the cAMP/PKA/CREB pathway, increase cAMP levels and increase BDNF levels, and improve neuronal survival after stroke injury. It has been reported that the representative drug roflumilast of PDE4 inhibitor is used to treat acute ischemic stroke (*Kwak et al., 2008*).

### PDE4 and synaptic remodeling in stroke injury

Signal transmission between neurons in the nervous system depends on the synaptic structure to achieve. Synaptic structure is a functional unit of receptors acting on neurotransmitters. The integrity of synaptic structures in the nervous system is the material basis for the physiological functions of neurons (*Burkhardt & Sprecher, 2017*). Brain tissue ischemia and hypoxia leads to cell energy depletion, reduction in ATP production, and a cascade amplification effect due to cAMP signaling pathway imbalance, resulting in a series of changes. Intracellular energy failure cannot maintain cell ion homeostasis. Intracellular acidosis and an increase in calcium content lead to excitatory toxicity and eventually result in cell death and apoptosis (*Chen et al., 2017a*). Neuronal apoptosis and death in the ischemic penumbra can lead to destruction of synaptic structures and disruption of signaling between neurons. Decreased neurotransmitter levels and disruption of the transmission pathway lead to impaired neuronal function, which in turn produces secondary brain damage that greatly threatens the patient's life and health (*Ito et al., 2006*).

The synaptic structure of the nervous system has plasticity, and the synapse exhibits corresponding changes in structure and function when the surrounding environment changes or is in a pathological state. The plasticity of neuronal synaptic structures is manifested in the neural circuit, synaptic morphology, *etc*. The indicators include synaptic interface curvature, synaptic size, synaptic number, synaptic length and spine density. The plasticity of neuronal function is reflected in behavioral characteristics, mental behavior and brain function, such as autonomous learning, long-term and short-term memory functions (*Lohmann & Kessels, 2014*). Synaptic remodeling is not only related to neurons in the nervous system, but also closely related to astrocyte and microglia (*Araque et al., 1999*; *Hynds et al., 2004*). In the early stage of cerebral ischemia, astrocytes and microglia are activated, and their cell morphology and function are also altered, further involved in the regulation of various pathways of nerve damage (*Liu, Tang & Feng, 2011*; *Gabryel et al., 2015*; *Choudhury & Ding, 2016*).

Current studies indicate that PDE4/ mitogen-activated protein kinase (MAPK)/ extracellular regulated protein kinases (ERK)/CREB and cAMP/PKA/CREB signaling pathways play important roles in neuronal synaptic remodeling and regulation of memory function. Moreover, PDE4 is closely related to these two pathways. PDE4 inhibitors can enhance long-term synaptic facilitation and enhance learning and memory processes by activating these two important pathways (*Waltereit & Weller, 2003*). Studies by *Li et al. (2011)* have demonstrated that the PDE4 inhibitor rolipram can significantly improve neuronal damage and learning and memory in SD rats with focal reperfusion injury. Other studies have also demonstrated that PDE4 inhibitors are involved in neural remodeling. They are related to the reduction of apoptosis through the regulation of apoptotic genes like bcl-2 and Bax, and the up-regulation of cAMP levels, which in turn enhances the function of neurotrophic factors (*Wang et al., 2012*).

### PDE4 and angiogenesis in stroke injury

Angiogenesis refers to the development of new blood vessels from existing capillaries or veins behind capillaries (*Carmeliet & Jain, 2011*). The primary purpose of generating new blood vessels in cerebral ischemic injury is to increase collateral circulation as a first-line defense on the ischemic side. Studies have shown that angiogenesis has been found in the ischemic penumbra of the brain and the animal model of stroke (*Krupinski et al., 1994*; *Zhang & Chopp, 2002*; *Hayashi et al., 2003*). Moreover, there is a positive correlation between the survival rate of patients and microvessel density, indicating that angiogenesis plays a crucial role in the recovery of cerebral ischemic injury (*Krupinski et al., 1994*). *Hu et al. (2016)* showed that the PDE4 inhibitor rolipram promoted the microvessel density in the cerebral ischemic border region and the increase in p-CREB expression in the ischemic penumbra through the cAMP/CREB pathway, indicating that rolipram can reduce cerebral ischemic damage by promoting angiogenesis (*Kumar et al., 2017*).

## CONCLUSION

In summary, after stroke, hypoxic-ischemic brain tissue occurs, leading to cell energy depletion, a reduction in ATP production and a cascade amplification effect due to an imbalance in the cAMP signaling pathway. This further induces a series of physiological and pathological changes in the body, such as energy failure, cell ion homeostasis disorders, acidosis, increased intracellular calcium content, excitotoxicity, increased cytotoxicity due to apoptosis, complement activation, blood-brain barrier (BBB) destruction, and glial cell activation. The key to alleviating stroke injury is to prevent the death of the ischemic penumbra neurons and the destruction of synapses after restoring blood flow in the brain tissue, and to reduce the secondary damage caused by stroke damage. PDE4 inhibitors have been demonstrated to reduce stroke damage by regulating cell-related pathways. These include reducing the inflammatory response, attenuating the cAMP signaling cascade, enhancing neurotrophic factors, promoting synaptic remodeling and angiogenesis, thereby playing a protective role in the tissue and cells of the ischemic penumbra and ultimately having an important role in the treatment of stroke damage.

Phosphodiesterase 4 provides a new drug target for the treatment of stroke, and its in-depth study will contribute to the further development of stroke treatment drugs.

## LIST OF ABBREVIATIONS

| | |
|---|---|
| **CVA** | cerebral vascular accident |
| **CS** | complete stroke |
| **RIND** | reversible ischemic neurological deficit |
| **TIA** | transient ischemic attack |
| **CEA** | carotid endarterectomy |
| **CAS** | carotid angioplasty and stenting |
| **tPA** | tissue plasminogen activator |
| **cGMP** | cyclic guanosine monophosphate |
| **cAMP** | cyclic adenosine monophosphate |
| **PDEs** | phosphodiesterase |
| **UCR1** | upstream conserved domains 1 |
| **UCR2** | upstream conserved domains 2 |
| **LR1** | linker region 1 |
| **LR2** | linker region 2 |
| **PKA** | phosphokinase-A |
| **AKAPs** | A-kinase anchoring proteins |
| **mAKAP** | A-kinase anchoring protein |
| **RACK1** | activated C kinase 1 |
| **CR3** | control region 3 |
| **CAMs** | cell adhesion molecules |
| **COPD** | chronic obstructive pulmonary disease |
| **TNF-α** | tumor necrosis factor |
| **SNP** | single nucleotide polymorphism |
| **HARBS** | high affinity rolipram binding state |
| **MAPK** | mitogen-activated protein kinase |
| **ERK** | extracellular regulated protein kinases |
| **RAC1** | Ras-related C3 botulinum toxin substrate |
| **ABI** | acquired brain injury |
| **CREB** | cAMP responsive element binding protein |
| **BDNF** | brain-derived neurotrophic factor |
| **NGF** | nerve growth factor |
| **CRE** | cAMP response element |
| **EPAC** | exchange protein activated by cAMP |
| **Akt** | protein kinase B |
| **AMP** | adenosine 5′-monophosphate |
| **AMPK** | adenosine 5′-monophosphate (AMP)-activated protein kinase |

### Funding

This study was supported by the Natural Science Foundation of China (Grant No. 82304434), GuangDong Basic and Applied Basic Research Foundation (Grant No. 2023A1515111199), China Postdoctoral Science Foundation (Grant No. 2022M713263), Public Welfare and Basic Research Project of Zhongshan City (Grant No. 2022B2015), and the Social Development Science and Technology Plan Project of Meizhou (Grant No. 2023B05, 2023B06). The funders had no role in study design, data collection and analysis, decision to publish, or preparation of the manuscript.

### Grant Disclosures

The following grant information was disclosed by the authors:
Natural Science Foundation of China: 82304434.
GuangDong Basic and Applied Basic Research Foundation: 2023A1515111199.
China Postdoctoral Science Foundation: 2022M713263.
Public Welfare and Basic Research Project of Zhongshan City: 2022B2015.
Social Development Science and Technology Plan Project of Meizhou: 2023B05 and 2023B06.

### Competing Interests

The authors declare that they have no competing interests.

### Author Contributions

- Jiahong Zhong conceived and designed the experiments, performed the experiments, analyzed the data, prepared figures and/or tables, and approved the final draft.
- Xihui Yu performed the experiments, prepared figures and/or tables, and approved the final draft.
- Zhuomiao Lin conceived and designed the experiments, performed the experiments, authored or reviewed drafts of the article, and approved the final draft.

### Data Availability

This is a literature review.

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
