# Peer review of "Phosphodiesterase 4 inhibition as a novel treatment for stroke"

_PeerJ, doi:10.7717/peerj.18905_

## Round 0.1 · original submission · Major Revisions

Two reviewers both suggested major revisions. For reviewer 2, please focus on comments 2 and 3. Reviewer 1's comments should be fully addressed.

·

Basic reporting

This is an interesting review that provides experimental support for a role of PDE4 inhibition in treating the consequences of stroke.

The authors should address the following points:

The authors mention that PDE4D is a risk gene for stroke. I don’t read the connection between having this risk gene and treating stroke patients with PDE4 inhibitors. How are these related? Or, is there no relation? This is not clear from the text and should be explained better.

Since the paper is focused on PDE4 inhibition for the treatment of stroke, the authors should be brief on current stroke treatments (section 2). There is too much detail that is not relevant for this review. This is not what a potential reader would be interested to read.

No reference is given for the following statement: “PDE4 is the most 238 abundant in brain tissue, reaching about 80%,…”. Please give reference for this claim. Further, the authors may choose to refer to the Lakics et al paper (https://doi.org/10.1016/j.neuropharm.2010.05.004), rather than referring to Gurney et al 2015.

Figure 1 is too simplistic. Why should Figure 2 be added? The review only relates to PDE

Although a comprehensive search was done, there are some review papers that are not cited by the authors (see below). Since these should be found with a thorough search, it can be questioned whether this search was optimal.
https://doi.org/10.1016/j.tips.2019.10.006 (highlighting the rational for using PDE4 inhibitors in brain disorders, including stroke).
https://doi.org/10.2174/1570159x17666191010103044 (providing a rational for using PDE4 inhibitors in acquired brain injuries, like stroke).

Minor:
The authors should make paragraphs in the text.

Experimental design

N.A.
Literature is not optimal

Validity of the findings

Can be questioned since literature search does not seem to be optimal

Additional comments

N.A.

Reviewer 2 ·

Basic reporting

no comment

Experimental design

no comment

Validity of the findings

no comment

Additional comments

In this manuscript, the authors summarize the relationship between inhibition of phosphodiesterase 4 and stroke. The author introduces the characteristics and treatment of stroke, and highlights the potential role of drug intervention. Subsequently, the author found that the comprehensive study of phosphodiesterase 4 as an innovative pharmacological target of stroke injury provided valuable insights for the development of therapeutic intervention in stroke treatment. Overall, it’s suggested that this manuscript could be published in PeerJ after revision.
1.Phosphodiesterase 4 inhibitors have been mentioned repeatedly in the manuscript. Please use a new chapter to describe the research status of phosphodiesterase 4 inhibitors in detail, and summarize their structure-activity relationship and structural optimization ideas. For example, by analyzing the co-crystal complex of compounds with the protein, the key amino acid residues to improve efficacy and selectivity were obtained.
2.Any compound that appears in the manuscript, please give the chemical structures of these compounds and give numbering according to the order in which they appear.
3.There are many PDE4 inhibitors entering clinical studies, such as AD-301 (NCT No. NCT02118766), Ensifentrine (NCT No. NCT04535986 or NCT04542057), AD-302 (NCT No. NCT02118792), etc., so please summarize the progress of clinical studies of these compounds clearly, what are the difficulties in the studies? and how the newly designed PDE4 inhibitors should overcome these problems?
4.In this manuscript, the structure of PDE protein is briefly introduced, and I suggest that the plane and three-dimensional structure of PDE4 protein be introduced in detail. What contribution do different domains make to the design of PDE4 selective small molecule inhibitors? What ideas can be provided for the development of new potent PDE4 inhibitors?
5.Ensure all references are up-to-date and relevant. Adding recent references could improve the manuscript's grounding in the current research landscape.
6.The grammar of this manuscript needs further refinement (including lines 52, 55, 61, 66, etc.), and it is recommended that someone well-versed in grammar revise it.

---

## Round 0.2 · accepted · Accept

Thank you - two reviewers have indicated their concerns have been met.

·

Basic reporting

The authors have adequately addressed my points

Experimental design

n.a.

Validity of the findings

OK

Reviewer 2 ·

Basic reporting

no comment

Experimental design

no comment

Validity of the findings

no comment

Additional comments

The author basically answered my question and agreed to accept the manuscript.